# A Methodology for Attributing the Role of Climate Change in Extreme Events: A Global Spectrally Nudged Storyline

Linda van Garderen[1], Frauke Feser[1], Theodore G. Shepherd[2]

[1]Institute for Coastal Research, Helmholtz-Zentrum Geesthacht, Max-Planck-Straße 1, 21502 Geesthacht, Germany
[2]Department of Meteorology, University of Reading, Reading RG6 6BB, United Kingdom

*Correspondence to*: Linda van Garderen (linda.vangarderen@hzg.de)

**Abstract.**

Extreme weather events are generally associated with unusual dynamical conditions, yet the signal-to-noise ratio of the dynamical aspects of climate change that are relevant to extremes appears to be small, and the nature of the change can be highly uncertain. On the other hand, the thermodynamic aspects of climate change are already largely apparent from observations, and are far more certain since they are anchored in agreed-upon physical understanding. The storyline method of extreme event attribution, which has been gaining traction in recent years, quantitatively estimates the magnitude of thermodynamic aspects of climate change, given the dynamical conditions. There are different ways of imposing the dynamical conditions. Here we present and evaluate a method where the dynamical conditions are enforced through global spectral nudging towards reanalysis data of the large-scale vorticity and divergence in the free atmosphere, leaving the lower atmosphere free to respond. We simulate the historical extreme weather event twice: first in the world as we know it, with the events occurring on a background of a changing climate, and second in a 'counterfactual' world, where the background is held fixed over the past century. We describe the methodology in detail, and present results for the European 2003 heatwave and the Russian 2010 heatwave as a proof of concept. These show that the conditional attribution can be performed with a high signal-to-noise ratio on daily timescales and at local spatial scales. Our methodology is thus potentially highly useful for realistic stress testing of resilience strategies for climate impacts, when coupled to an impact model.

## 1. Introduction

There is increasing interest in understanding and quantifying the impact of climate change on individual extreme weather and climate events. This is to be distinguished from detecting the effect of climate change on the statistics of extreme events (SREX, 2012). In the most commonly-used approach, changes in the probability distribution of an event class, whose definition is motivated by an historical event, are calculated by simulating large ensembles with an atmosphere-only climate model (Watanabe et al., 2013). The changes are computed between the 'factual' ensemble, corresponding to observed forcings (e.g. sea-surface temperatures (SSTs) and greenhouse-gas (GHG) concentrations), and a 'counter-factual' ensemble, corresponding to an imagined world without climate change. The latter is usually constructed by removing an estimate of the forced changes

in SSTs, and imposing pre-industrial GHG concentrations. As discussed by Shepherd (2016), this probabilistic approach has two prominent limitations. The first is that every extreme event is unique, but the construction of a general event class blurs the connection to the actual event and makes it difficult to link the event attribution to climate impacts. This is important because extreme impacts are not always associated with extreme meteorology (van der Wiel et al., 2020). The second limitation is that extreme events are generally associated with extreme dynamical conditions, and there is little understanding, let alone

agreement, on how those dynamical conditions might respond to climate change (Hoskins and Woollings, 2015; Shepherd, 2014). This represents an uncertainty in the probabilistic estimates that is difficult to quantify.

On the other hand, thermodynamic aspects of climate change such as warming and increasing specific humidity are robust in sign, anchored in agreed-upon physical understanding, and clearly emerging in observations (IPC, 2018). Moreover in many

cases the signal-to-noise ratio of the forced dynamical changes appears likely to be small (Deser et al., 2016; Schneider et al., 2012). Thus, although dynamical and thermodynamic processes are interwoven in the real climate system, it can be useful to regard the *uncertainties* in their forced response to climate change as being separable, at least to a first approximation. This has been a growing theme in climate change attribution over the past few decades. The distinction between thermodynamic and dynamical changes is not precise, and various ways of implementing the separation diagnostically have been used in

different contexts. For extratropical regional climate, it has been common to regard the component of change congruent with large-scale internal variability (e.g. as defined by Empirical Orthogonal Functions or by Self-Organizing Maps) as 'dynamical' (Deser et al., 2016; Horton et al., 2015), and the residual as 'thermodynamic'. For tropical climate or for extratropical storms, dynamical changes are instead commonly identified with changes in vertical velocity (Bony et al., 2013; Pfahl et al., 2017). In the absence of evidence to the contrary, a reasonable hypothesis is that the forced dynamical changes are undetectable; this

hypothesis is implemented explicitly in the 'pseudo global warming' methodology used for regional climate studies (Schär et al., 1996), and in the 'dynamical adjustment' methodology used to study observed climate trends (Wallace et al., 2012).

Trenberth et al. (2015) suggested that the same thinking could be usefully applied to the attribution of individual extreme events. Specifically, the extreme dynamical circumstances leading to the event could be regarded as given, i.e. arising by

chance, and the question posed of how the event was modified by the known thermodynamic aspects of climate change. This conditional framing of the attribution question was subsequently dubbed the 'storyline' approach (Shepherd, 2016), and has a precedent in the application of dynamical adjustment to extreme seasonal climate anomalies (Cattiaux et al., 2010). As emphasized by Shepherd (2016) and NAS (2016), there is actually a continuum between the storyline and probabilistic approaches: storylines are highly conditioned probabilities, and probabilistic approaches generally involve some form of

dynamical conditioning too, through the imposed SST patterns. However, the extent of conditioning imposed by constraining the atmospheric state is so severe that in practice the storyline approach can be regarded as deterministic, just as weather forecasts, whilst probabilistic in principle, are interpreted deterministically when the ensemble spread is sufficiently narrow.

By focusing on the known effects of climate change, the storyline approach seeks to avoid 'Type 2' errors or missed warnings, in contrast to the probabilistic approach which, by needing to reject the null hypothesis of no climate change whatsoever, seeks to avoid 'Type 1' errors or false alarms (Lloyd and Oreskes, 2018; Trenberth et al., 2015). A colloquial way of putting this is that rather than asking what extreme events can tell us about climate change, we ask what known aspects of climate change can tell us about particular extreme events. Although its results are not expressed probabilistically, the storyline approach enables a quantitative estimate of climate change with a clear causal interpretation (Pearl and Mackenzie, 2018). Notwithstanding the need for asking both kinds of questions, as they provide different kinds of information (Lloyd and Shepherd, 2020), the storyline approach is a new development and there are as yet not so many studies employing this approach.

In previous applications of the storyline approach, individual extreme weather events have been dynamically constrained through boundary conditions applied to a regional model (Meredith et al., 2015) or by controlling the initial conditions in a weather forecast model (Patricola and Wehner, 2018). More recently, nudging the free atmosphere to reanalysis data (leaving the boundary layer free to respond) has been applied in a global medium-resolution atmospheric model to constrain the dynamical conditions leading to heat waves, first to determine the effect of soil moisture changes on selected recent heat waves (Wehrli et al., 2019), and subsequently to determine the effect of past and projected future warming on the 2018 Northern Hemisphere heatwave (Wehrli et al., 2020). The concept of nudging the atmospheric circulation in order to impose the dynamical conditions has a long history. In particular, spectral nudging (von Storch et al., 2000; Waldron et al., 1996) allows for scale-selective nudging so that only the large spatial scales of the model are constrained, while the smaller scales, including those relevant to extreme events, are free to be simulated by the high-resolution model. The climate model can thus potentially add value and regional detail to the coarser-resolution forcing data set. Spectral nudging has been used in regional climate modelling (Feser and Barcikowska, 2012; Scinocca et al., 2015) and in boundary-layer sensitivity studies (van Niekerk et al., 2016). Note that in all these modelling approaches, the dynamical constraint is imposed 'remotely' from the phenomenon of interest (in space, time, and/or spatial scale), in contrast to the diagnostic approaches mentioned earlier, and thus preserves the physical interplay between dynamics and thermodynamics within the extreme event itself.

The purpose of this paper is to provide a methodological underpinning for the application of large-scale spectral nudging of divergence and vorticity in a global high-resolution atmospheric model, for the purpose of attributing the role of thermodynamic aspects of climate change (or other conditional perturbations) in extreme events of various types and timescales. A key question is to determine what level of refinement of the attribution, in both space and time, is possible. The outline of the paper is as follows. In section two, we elaborate on the technicalities of spectral nudging within the ECHAM6 model and its parameter sensitivities, as well as the construction of the counterfactual simulations. In section three, we exemplify the method by applying it to two well-studied heatwaves: the European 2003 heatwave, and the Russian 2010

heatwave. As well as identifying some important differences between the two events, we examine the signal-to-noise ratio of our attribution. A concluding discussion follows in section 4.

## 2 Method

### 2.1 Spectral Nudging

The spectral nudging technique is well established within the context of regional climate modelling (Miguez-Macho et al., 2004; von Storch et al., 2018; von Storch et al., 2000; Waldron et al., 1996). In this approach, so-called 'nudging terms' are added to the large-scale part of the climate model trajectory, which draw the model towards reanalysis data. Global spectral nudging (Kim and Hong, 2012; Schubert-Frisius et al., 2017; Yoshimura and Kanamitsu, 2008) works in a similar way. It constrains large-scale weather patterns of the climate model, such as high and low pressure systems or fronts, to stay close to reanalysis data in order to derive a global high-resolution weather reconstruction. The general idea is that the realistic large-scale state of the reanalysis data is followed by the GCM, while at smaller scales the model provides additional detail to improve high-resolution weather patterns. Another merit of the approach is the potential to reduce inhomogeneities in the data set by using only a very limited number of variables from the reanalysis data, although this is less of an issue for our application because we compare factual and counter-factual simulations for the same large-scale conditions, so any inhomogeneity in the reanalysis would apply equally to both. For the same reason, our approach can be expected to be robust to any differences between reanalyses. In order to define a noise level for our analysis, we construct small ensembles of three factual and three counter-factual simulations. Although such small ensembles are clearly inadequate for quantifying conditional probabilities, they have been successfully used in the past (e.g. Shepherd, 2008) to identify robust differences between the two ensembles from a deterministic perspective, which is our interest here.

### 2.2 ECHAM6 application

For this study, we use the high-resolution T255L95 GCM ECHAM6 (Stevens et al., 2013) with the JSBACH land component sub-model (Reick et al., 2013), however the method is applicable to any atmospheric GCM. SSTs and SICs are prescribed from NCEP1 reanalysis data (Kalnay et al., 1996). ECHAM6 is globally spectrally nudged towards the NCEP1 reanalysis data to achieve realistic weather patterns and extreme events of the past. However, any other reanalysis should provide similar results, since only the large-scale fields are nudged. We chose NCEP1 due to its starting date in 1948, which is earlier than any of the other reanalysis data, enabling application of our method over a longer period of time. It is conceivable that for certain kinds of extreme events involving a tight coupling between resolved and parameterized processes, ensuring consistency between the reanalysis and the model would be beneficial. In a previous application nudging was applied for pressure, temperature, vorticity and divergence (Jeuken et al., 1996) with a constant height profile throughout the entire atmosphere. However, we want to reproduce only the large-scale atmospheric circulation, and in particular leave the thermodynamic fields

(temperature and moisture) free to respond, hence we only nudge vorticity and divergence in the free atmosphere. The aim is to constrain the model as little as possible so that it can freely develop small-scale meteorological processes and extreme events, while still achieving an effective control of the large-scale weather situation.

The nudging of variable X over time is applied in the spectral domain as follows (adapted from Jeuken et al. 1996):

$$\frac{\partial X}{\partial t} = \begin{cases} F_X + G(X_{NCEP} - X) \ for \quad n \leq 20 \,, p < 750 hPa \\ F_X \hspace{4.2cm} otherwise \end{cases} \qquad (1)$$

where $X$ is the variable to be nudged (either vorticity or divergence), $F_X$ is the model tendency for variable $X$, and $X_{NCEP}$ is the state of that variable in NCEP1. The thresholds $p$ and $n$ need to be met for nudging to happen, namely pressure $p$ must be below 750 hPa, and the spherical harmonic index $n$ must not exceed 20. $G$ is the relaxation coefficient in units of $10^{-5}\,s^{-1}$ determining the nudging strength. Nudging is performed at every time step.

We applied most settings according to Schubert-Frisius et al. (2017), including the usage of spectral nudging in both meridional and zonal directions. We use a plateau nudging-strength height profile (see Figure 1a), which starts at 750hPa, then quickly increases up to its maximum nudging strength, stays there for higher tropospheric and lower and medium stratospheric levels until it again quickly tapers back to zero at a height corresponding to 5 hPa. The reason for the latter choice is that above 5 hPa there is no NCEP1 reanalysis data available.

The strength of nudging is determined by the relaxation coefficient (G, in $10^{-5}\,s^{-1}$), see Equation 1. The relaxation coefficient is often described using the e-folding time ($G^{-1}$, in $10^5\,s$) which represents the simulated time necessary for nudging to dampen out a model-introduced disturbance. For example, if the e-folding time is 10 hours then the nudged model will dampen out that disturbance (with an assumed amplitude of 1) to a value of 1/e and thus greatly reduce it within 10 hours. A larger relaxation coefficient implies a stronger nudging and translates into a shorter e-folding time or dampening time (von Storch et al., 2000). We have tested several e-folding times to see if the settings could be further relaxed and still reproduce the large-scale weather conditions. In Figure 1b the impact of the tested e-folding time settings on the temporal evolution of the two-meter temperature averaged over Europe (10°W-30°E, 35-60°N) in comparison to ERA-Interim is shown through November 2013. There is little difference visible between the 50-minute and 5-hour e-folding times. The 10-hour results start to show small deviations, whilst the 20-hour results deviate even more noticeably. On the basis of this sensitivity study, we conclude that the e-folding time can safely be relaxed from 50 minutes to 5 hours without losing the accuracy of the results.

We similarly aim to limit the range of spatial scales being nudged as much as possible. In Figure 1c we show the two-meter temperature results for the different nudging wavelengths in comparison to ERA-Interim. The original T30 settings used by

Schubert-Frisius et al. (2017), which translate to a minimum wavelength of approximately 1300 km (360°/30*111 km), show comparable results to the T25 and T20 resolutions. The nudging was therefore relaxed to the T20 resolution, which translates to a minimum wavelength of approximately 2000 km (360°/20*111 km). This should be sufficient to resolve the large-scale circulation while allowing smaller-scale processes, related to local weather events, to develop freely. In Figure 2 the geopotential height anomalies for summer 2010 in the factual and counterfactual simulations show a strong resemblance. Even though the background conditions of the two simulations are different (which is further explained in section 2.3), the blocking pattern formed over Russia in 2010 is clearly present in both simulations, demonstrating the capability of our nudging method to reproduce the complex dynamical situation.

We used ECHAM_SN throughout this paper to calculate climatological data for comparison to our own findings. The ECHAM_SN dataset is a spectrally nudged global historical simulation from 1948-2015 (Schubert-Frisius et al., 2017). It nudged vorticity and divergence towards NCEP1 in a vertical plateau shaped profile, equal to the profile we use, at spatial scales corresponding to T30 or larger, with an e-folding time of 50 minutes.

**2.3 Simulating the Counterfactual**

In this study, as in probabilistic event attribution, counterfactual and factual climate simulations are used to assess the effect of climate change on extreme events. Factual is defined as the world as we know it, or a historical simulation. Counterfactual is defined as an imagined modern world without climate change. In our simulations, land-use and volcanic activity, as well as aerosol forcing and sea ice concentration, are unchanged between factual and counterfactual. The differences between the two worlds are created by altering two important aspects of the simulation: a) Sea Surface Temperature (SST) and b) Greenhouse Gases (GHG). Both worlds are spectrally nudged in the same way. A potential way to check the results of the counterfactual simulation, especially for simulations over a longer time span, is to study the consistency between the inferred signals of climate change for smaller climate forcings (e.g. since mid-century) and the attributed changes in the observational record. Our simulations are five years each and therefore cannot be tested in this way. However for longer simulations such a test would be beneficial.

SST patterns such as the Atlantic Multidecadal Oscillation or El Niño greatly influence weather extremes. Therefore, as with probabilistic event attribution, we impose the same SST variability for both the factual and counterfactual simulation, based on the observed SST pattern. (However, this is expected to be less critical in our case since we are imposing the large-scale atmospheric circulation.) We create the counterfactual SST conditions by subtracting a climatological warming pattern from the observed pattern, which is a standard procedure in probabilistic event attribution studies (Otto, 2017; Vautard et al., 2016; Stott et al., 2016). Although it is common to consider different climatological warming patterns as a means of exploring uncertainty, this is not so relevant in our case since the large-scale circulation is imposed. The climatological warming pattern

is computed using the ECHAM6 CMIP6 (MPI-ESM1.2-HR) control and historical simulations at an atmospheric resolution of T127 (Müller et al., 2018). The procedure is shown in Equation 2:

$$SST_{t,c} = SST_t^{NCEP1} - \left(SST_{t,h}^{CMIP6} - SST_{t,pi}^{CMIP6}\right) \quad (2)$$

where $SST_{t,c}$ is the counterfactual SST at time $t$, $SST_t^{NCEP1}$ is the NCEP1 SST at time $t$, $SST_{t,h}^{CMIP6}$ is the CMIP6 historical SST at time $t$, and $SST_{t,pi}^{CMIP6}$ is the CMIP6 pre-industrial SST at time $t$ (for the latter, the only relevant time dependence would be

seasonal). In our present implementation, which targets boreal summer only and concerns only a fairly short time period, the seasonal time-dependence is suppressed and the historical CMIP SST's are taken to be the 2000-2009 average. For a simulation covering a full year the warming pattern should be made seasonal, and for one covering several decades it would furthermore need to be weighted over time. In Figure 3 the CMIP6 SST warming pattern shows a good resemblance to the observed HadSST3 warming pattern. The HadSST3 pattern is obtained by subtracting the 1880-1890 average from the 1980-1990

average SST values. The general warming and cooling patches in the Pacific Ocean and Atlantic Ocean south of Greenland agree well. Also, the warming north of Scandinavia is clearly visible in both warming patterns. Despite the observational data-void region east of Greenland and north of Iceland, there is a good resemblance of our modelled warming pattern with observations. Note that pre-industrial SST observations were dependent upon ship records which in the polar region were very few (Rayner et al., 2006), causing this part of the observational data set to be incomplete.

For technical reasons, we did not alter the sea ice concentration (SIC) in the counterfactual simulations. Given that the atmospheric circulation is nudged, changes in SIC are not expected to be relevant for summertime heatwaves, as Arctic amplification from sea ice loss is a wintertime phenomenon (Screen and Simmonds, 2010). In Figure 4 the counterfactual SSTs for July 2003 and July 2010 are shown together with the factual SIC. This shows that the sea-ice edge is well away from

the European and western Russian domains. Moreover, even under counterfactual conditions the SST remains almost completely physically self-consistent with the SIC according to the constraints of Hurrell et al. (2008); in particular, there are only a very few isolated regions where the SST falls below -2°C. Nevertheless, we tested the impact of altering SIC in a counterfactual simulation of the Russian heatwave based on the counterfactual SST's, using the linear relation found by Hurrell et al. (2008). Specifically, SIC was set to 100% for SST's below -1.7°C, and to 0% for SST's above 3°C, with a linear

interpolation in between. The results show no differences compared to the unaltered SIC counterfactual members (see Figure 5b). However, to apply our method to other seasons or regions in close proximity to areas of sea ice loss, the counterfactual simulations would benefit from including SIC changes in the same way as was done with SST.

In the factual simulation the GHGs change according to observed values (Meinshausen et al., 2011). In the counterfactual

simulation, GHGs remain at their 1890 values as listed in Table 1. This means that, strictly speaking, our attribution is to the

combined effects of anthropogenic climate change (including aerosol forcing) recorded in the SSTs, as well as the direct radiative effects of GHG forcing.

The default initial atmospheric state of the ECHAM6 model is a random state during the simulated mid 1990's. Changing that initial state to a counterfactual initial state requires a spin-up time, to allow the atmosphere and land surface enough time to reach a new equilibrium state with their new boundary conditions. To accomplish this we run a non-nudged counterfactual spin-up ensemble for three model years with three members. We chose a three year spin-up after confirming the soil moisture was adapted to the new counterfactual situation (not shown). Each member was initiated at a different starting date (January, February, March 1995). The results of these spin-ups are three random atmospheric counterfactual states, which are used as initial conditions for the counterfactual experiments. Although in principle both the factual and counterfactual conditions define conditional probabilities, our three-member ensembles are certainly not sufficient to estimate those probabilities. As noted earlier, our goal here is simply to determine the robustness of the deterministic differences between the factual and counterfactual ensembles. The ECHAM_SN simulation and the altered SIC simulation provide out-of-sample tests of robustness for the factual and counterfactual ensembles, respectively. Figure 5 shows that in both cases, these simulations fall largely within the range of the three-member ensembles.

For the European 2003 heatwave the three counterfactual members run from 1 March and are initialized with the spin-up counterfactual atmospheric state members (year three, March). The three factual members are started one month apart from each other (in January, February and March 2003), initialized with the corresponding atmospheric state from the ECHAM_SN data set. For the Russian 2010 heatwave the three counterfactual members run instead from 1 January, because of the known importance of soil preconditioning for this event (Wehrli et al., 2019). The three factual members again run with one-month differences in their starting dates, but here from November 2009, December 2009, and January 2010, again initialized with the corresponding state from the ECHAM_SN dataset. For analysis regions we select 10°W-25°E/35-50°N as the domain for the European heatwave 2003, and 35-55°E/50-60°N for the Russian heatwave 2010, in line with previous literature (Dole et al., 2011; García-Herrera et al., 2010; Otto et al., 2012; Rasmijn et al., 2018; Wehrli et al., 2019).

For the summer of 2003, the global temperature difference between factual and counterfactual simulations is 0.64°C, while for the summer of 2010 the difference is 0.66°C. From observations we know that the earth has experienced a global warming of approximately 0.7–0.8°C between preindustrial times and 2010 (IPCC, 2018). Our modelled global warming, found through the difference between the factual and counterfactual simulations, thus represents this difference well albeit with a slight underestimation.

## 3. Results

To illustrate our method, we provide two examples, namely the European heat wave of 2003 and the Russian heat wave of 2010. These events are considered the two strongest European heatwaves on record (Russo et al., 2015; Russo et al., 2014). In section 3.3 we look deeper into the signal-to-noise ratio of each of the examples and how they compare to each other.

### 3.1 European Heatwave 2003

The European summer of 2003 was exceptionally hot and exceptionally dry (Black et al., 2004; Schär et al., 2004; Stott et al., 2004). Two heatwaves occurred, a milder one in June and an extreme heatwave in August, with peak temperatures in France and Switzerland (Black et al., 2004; Schär et al., 2004; Trigo et al., 2005) but also affecting Portugal, northern Italy, western Germany and the UK (Feudale and Shukla, 2011a; Muthers et al., 2017). Temperatures exceeded the 1961-1990 average by 2.3–12.5°C, depending on location, without much cooling during the night (García-Herrera et al., 2010; Schär et al., 2004; Stott et al., 2004; Muthers et al., 2017). The 2003 summer was at that point in time not just the hottest on record (Bastos et al., 2014; Fink et al., 2004), it was the hottest summer in the past 500 years (Luterbacher et al., 2004). The consequences were devastating. Estimates account for 22,000–40,000 heat-related deaths, $12-14 billion in economic losses, 20-30% decrease of Net Primary Productivity (NPP), 5-10% of Alpine glacier loss and many more human health related issues due to increased surface ozone concentrations (Ciais et al., 2005; Fischer et al., 2007; García-Herrera et al., 2010).

Both the June and August heatwaves were caused by stationary anticyclonic circulations, or blocking (Black et al., 2004). The first block formed in June, broke and quickly reformed in July which then caused the second heatwave in August (García-Herrera et al., 2010). However, the extreme temperatures cannot be explained by atmospheric blocking alone. Due to large precipitation deficits in spring that year, the heatwaves happened in very dry conditions. The lack of clouds and soil moisture caused latent heat transfer to turn into sensible heat transfer, which dramatically increased surface temperatures (Bastos et al., 2014; Ciais et al., 2005; Fischer et al., 2007; Fink et al., 2004; Miralles et al., 2014). It is considered highly unlikely that the 2003 European heatwaves would have reached the temperatures they did without climate change (Hannart et al., 2016; Schär et al., 2004; Stott et al., 2004). The probabilistic event attribution studies show an increased likelihood of the extreme temperatures from increased GHGs (Hannart et al., 2016; Schär et al., 2004; Stott et al., 2004). Other studies focused on the exceptionally high SSTs in the Mediterranean Sea and North Sea as a cause of reduced baroclinicity, providing an environment conducive to blocking (Black et al., 2004; Feudale and Shukla, 2011a, b). By applying the storyline approach, we can consider both causal factors together and shed some additional insight on this event. The dry spring leading up to the warm summer conditions was captured by initializing the simulations by 1 March at the latest.

In Figure 5a, the daily evolution of the domain-averaged temperature at two-meter height for June, July and August for each of the ensemble members is plotted in comparison to the ECHAM_SN 5th-95th percentile (1985-2005) climatology and ERA-

Interim (Dee et al., 2011). The ECHAM_SN 2003 temperature is also plotted for reference, and shows a strong coherence with the factual ensemble, confirming the appropriateness of using the ECHAM_SN climatology as a reference for our factual simulations. The first thing to note is that the factual and counterfactual ensembles evolve very similarly in time but (except for the third week of June) are well separated, by approximately 0.6°C, indicating a high signal-to-noise ratio at daily resolution for the domain average. This value of 0.6°C is in line with the global mean warming. ERA-Interim and the factual members show a strong correlation in time, although the ERA-Interim temperatures are higher especially in June and during the heatwave in the first half of August. The factual temperatures exceed the 95th percentile several times in June, July and August. In August, the exceedance lasts for almost two weeks whereas in June it does so for approximately one week. The counterfactual temperatures are not quite so extreme; they exceed the 95th percentile only for a few days at a time in June and August. Nevertheless, it is clear that there would have been a European heatwave in 2003 even without climate change, albeit with less extreme temperatures. This analysis thus supports both of the perspectives on the event discussed earlier, whilst providing a daily resolution of the climate-change attribution.

The temperature differences between the factual and counterfactual ensembles are spatially nonuniform over Europe. In Figure 6a the factual members' average of the two-meter temperature and geopotential height (z500) show the meteorological situation averaged over half-month periods following García-Herrera et al. (2010). Figure 6b shows the local differences in two-meter temperatures between the counterfactual and factual ensemble averages. Stippling is added to each grid point where all the three factual members are at least 0.1°C warmer than all the counterfactual members. There is strong local variance, especially during the heatwave in the first half of August, with differences of up to 2.5°C. In the first period (1-15 July) the local differences are generally modest, except in northern Spain where they reach 1.5-2°C. In the second and third half-month periods (16-31 July, 1-15 August), the temperatures in the factual simulations can locally be up to 2-2.5°C higher than in the counterfactual simulations, with the differences spread over a large area including Spain, Portugal, France, Germany, Hungary and Romania. During the period 1-15 August, which according to Figure 5a was the peak of the heat wave, the hottest area in Europe (Figure 6a) is located in south-west France and southern Iberia. However the largest differences between the factual and counterfactual simulations (up to 2.5°C) are found to the north of both of these regions, suggesting a shift of the peak temperature. In the second half of August, there are still some strong temperature differences visible over most of these regions, although the differences over western France have dampened.

As noted earlier, the dryness of the soil has been identified as an important contributing factor to the 2003 heatwave. Our interest here, however, is on whether the soil wetness differed between factual and counterfactual conditions. In Figure 7a we see a very similar decline in soil wetness for both the factual and counterfactual ensemble members from May until the end of August. The counterfactual simulations start out with somewhat higher soil wetness than the factual simulations, but over the

course of the summer the values of both sets of simulations move closer towards each other, so that by August the ensembles are close together. Thus it does not appear that climate change had a first-order impact on soil wetness in this case.

## 3.2 Russian Heatwave 2010

In August 2010 western Russia was hit by an unprecedented heatwave caused by a large quasi-stationary anticyclonic
circulation, or blocking (Galarneau et al., 2012; Grumm, 2011; Matsueda, 2011). It was a heatwave that broke all records such as temperature anomalies during both day and night, temporal duration, and spatial extent. The effect of soil wetness, or rather the lack thereof, on the magnitude of the temperatures was profound (Lau and Kim, 2012; Rasmijn et al., 2018; Wehrli et al., 2019; Bastos et al., 2014). The 2010 Russian heatwave is considered the most extreme heatwave in Europe on record (Russo et al., 2015). Approximately 50,000 lives were lost, 5,000 km$^2$ forest burned, 25% of the crop failed and over 15 billion US
dollars' worth of economic damage was recorded due to this heatwave (Barriopedro et al., 2011; Lau and Kim, 2012; Otto et al., 2012; Rasmijn et al., 2018). In some of the attribution studies, the heatwave was primarily attributed to internal variability as the dynamical situation strongly depended on the El Niño Southern Oscillation (ENSO) being in a La Niña state (Dole et al., 2011; Russo et al., 2014; Schneidereit et al., 2012). However, the likelihood of the temperatures reaching such extreme values has also been assessed as being significantly exacerbated by climate change (Otto et al., 2012; Rahmstorf and Coumou,
2011) . As with the previous example, the storyline approach can represent both of these perspectives. Moreover, it overcomes the limitation that the climate models used to perform probabilistic event attribution generally have trouble reproducing a blocking situation correctly (Trenberth and Fasullo, 2012; Watanabe et al., 2013).

In Figure 5b, the daily evolution of the domain-averaged temperature at two-meter height for each of the ensemble members
is shown in comparison to ECHAM_SN 2010, the ECHAM_SN 5th-95th percentile climatological temperatures (1985-2015) and ERA-Interim. ERA-Interim temperatures correlate highly with the counterfactual members, though are somewhat higher at the end of June and beginning of July, and decline much more rapidly following the heatwave halfway through August. Starting after the second half of July both the factual and counterfactual temperatures exceed the 95th percentile climatological temperature, peak around the 8th of August and return to climatological temperatures around the 17th of August. This analysis
shows that this would have been an unprecedented event, even without climate change. The differences between the factual and counterfactual temperatures during the core of the heat wave are noticeably higher (about 2°C) than in the European heatwave 2003, as is the spread between the ensemble members. In contrast to the European case, the anthropogenic warming during the core of the heat wave is considerably higher than the global-mean warming. We attribute both aspects — the greatly enhanced anthropogenic warming, and the larger internal variability — to the fact that the Russian domain is much further
inland than the European domain, and thus the blocking conditions cut off the influence of the SST forcing and allow a direct radiative effect of GHG increases (Wehrli et al., 2019). Note that western Russia is known for having large internal variability (Dole et al., 2011; Russo et al., 2014; Schneidereit et al., 2012), which is clearly apparent in our results. It is also the case that

the Russian domain is smaller than the European domain by a factor of 3.4, which would furthermore tend to increase the variability in the domain-averaged temperature shown in Figure 5.


The range of temperature differences between factual and counterfactual simulations reach values up to 4°C locally, as seen in Figure 6d. Note that the scale for the Russian heatwave reaches up to 4°C, whereas the scale for the European heatwave reaches only 2.5°C. In the first half-month period (1-15 July), when the heatwave had not yet started, the local temperature differences are between 0.5-2.5°C, with the maximum differences in the south-east of the domain. The temperature differences are largest

in the core of the block region, reaching up to 3.5°C in the south-east in the second period (16-31 July) and up to 4°C in the south, below Moscow, in the third period (1-15 August). The block broke in the fourth period (16-31 August) and resulted in a virtual elimination of the temperature difference. In contrast to the European heatwave 2003, here the biggest temperature differences between factual and counterfactual are found in the regions with the highest temperatures.

As with the European heatwave 2003, the differences in soil wetness do not appear to be of first-order importance to explain the temperature differences between the factual and counterfactual simulations. In Figure 7b the soil wetness in the factual simulations is seen to decrease somewhat more rapidly than in the counterfactual, which could be due to the higher surface temperature and thus greater evaporation of soil moisture. However, the soil wetness values are overlapping, and even cross each other in the beginning of August. These findings are in agreement with those of Hauser et al. (2016), who reproduced the

Russian heatwave under 1960 conditions and found that the dry conditions occurred there too, thus concluding they found no direct link between the drought conditions and climate change. It must be emphasized that this is not to downplay in any way the impact of soil wetness on the event itself, which has been well established in the literature. It is only to indicate that the impact would have been there even without climate change.

### 3.3 Signal-to-Noise Analysis

The temperature differences found between the factual and counterfactual simulations are meaningful if they are outside of the internal variability within each ensemble. A different way of saying this is that the differences are meaningful if the two ensembles are distinguishable from each other. To assess this in a statistical manner, temperature differences between pairs of factual members (FF), pairs of counterfactual members (CC), and factual-counterfactual pairs (FC) are plotted for each half-month period in Figure 8. The FF and CC pairs have a median close to zero and represent the noise level; in both cases there

are three pairs (F1-F2, F2-F3, F3-F1 / C1-C2, C2-C3, C3-C1). The FC pairs contain the signal; here there are nine pairs (F1-C1, F2-C2, F3-C3, F2-C1, F3-C2, F1-C3, F3-C1, F1-C2, F2-C3). Each box plot represents the distribution of two-metre temperature differences across the pairs and across all grid points. The half-monthly panels represent distributions of half-month averaged values, and the daily panels distributions of daily values within the half-month period.

The daily differences for the European heatwave (Figure 8a) show a median value of about 0.6°C, irrespective of whether the timeframe is during the heatwave itself, directly before or directly after it, consistently with Figure 5a. Although these are not really probability distributions (since they include contributions from different locations within the domain), we can use the inter-quartile ranges as measures of signal and noise. The median difference for FC is above the 75th percentile of both CC and FF for daily values, giving confidence that our results are clearly above the noise level. Half-monthly time averages (Figure

8b) produce nearly identical median values, but we see that the spread is much smaller, as expected. The 25th percentile of FC now lies above the 75th percentile of the CC and FF boxes.

The differences between CF and either FF or CC for the Russian heatwave (Figure 8c,d) are clearly larger than for the European heatwave, and in contrast to the European case vary substantially between the different periods. Consistently with Figure 5b,

in the periods outside of the core of the heatwave (1-15 July; 16-31 August) the median difference between FC is about 1°C. Inside the core heatwave period (16-31 July; 1-15 August), however, the median difference is more like 2°C, reaching 2.2°C for 1-15 August. During this latter period the 5th percentile whisker of half-monthly FC differences is above the 75th percentile of FF and CC, which is a very strong signal indeed. When looking at the results for individual members the larger internal variability within the Russian domain (apparent also in Figure 5b) is clearly visible (not shown), as compared with the

European case.

## 4. Discussion and Conclusion

We have presented a detailed description and assessment of a global spectrally nudged storyline methodology to quantify the role of known thermodynamic aspects of climate change in specific extreme weather events. In this methodology, the particular dynamical conditions leading to the event are taken as given, i.e. are regarded as random, and the attribution is therefore highly

conditional. Thus, as with all such storyline approaches to extreme event attribution, the effect of climate change on the occurrence likelihood of those dynamical conditions is not assessed. In that respect, this approach is complementary to the more widely-used probabilistic event attribution. However, since most results of probabilistic event attribution appeal in any case to the known thermodynamic aspects of climate change, it can be argued that not much is lost in the storyline approach, yet much is gained by the specificity. This is especially the case for extreme events whose dynamical conditions are not well

represented in climate models, e.g. blocking. Spectral nudging enables the reproduction of extreme events with their particular dynamical details, allowing the same dynamical events to be reproduced in simulations with different boundary conditions, and thereby achieving a high signal-to-noise ratio of the climate change effect. The combination of both methods — global spectral nudging and the storyline method — thus presents a way to quantify, in great detail, the role of known thermodynamic aspects of climate change, together with the specific dynamical conditions, in selected extreme events which happened in the

recent past. This can help reconcile the sometimes different perspectives on those events that appear in the literature (some emphasizing climate change, others emphasizing internal variability).

We illustrated the method by applying it to two extreme events that have been the subject of much study: the European heatwave of 2003, and the Russian heatwave of 2010. By using a small ensemble of both factual and counterfactual simulations, we were able to determine a noise level for our analysis. This revealed that the (conditional) signal of climate change is determinable at both daily timescales and local spatial scales. It follows that our methodology could be used to drive climate impact models, and thus perform realistic stress-testing of resilience strategies. With regard to the two heatwave examples, our analysis revealed a striking contrast between the two events. In the European heatwave of 2003, the effect of climate change was to increase temperatures across Europe by about the global-mean warming level throughout the summer, and the heat wave was simply the dynamical event riding on top of that. In the Russian heatwave of 2010, in contrast, the effect of climate change was much higher than the global warming level, and was particularly enhanced, by approximately three-fold, during the peak of the heatwave. We interpret this difference as reflecting the role of direct GHG radiative forcing, which can become apparent when air masses are cut off from marine influence. However, further analysis would be required to confirm this hypothesis.

It is not possible to make a direct comparison between our results and probabilistic attribution of these heat waves, because they are answering different questions, and the conditionalities are quite different. However, from a methodological perspective it is useful to contrast the *nature* of the attribution statements that can be made using the different methods. We do this in Table 2 for the case of the Russian 2010 heat wave. Having said that, there is a continuum between storyline and probabilistic approaches (Shepherd, 2016), and it is possible to imagine intermediate set-ups which would provide a seamless connection between event attribution and probabilistic weather prediction (NAS, 2016). These would need to involve large ensembles (to calculate conditional probabilities), and pay more attention to the self-consistency of how the counterfactual conditions are imposed. An example is the recent use of an operational subseasonal-to-seasonal prediction system, which involves modifying the atmospheric state and land-surface conditions as well as the SSTs in generating the counterfactual (Wang et al., 2020).

The nudged global storyline method is an important step towards a holistic approach within the attribution of individual extreme events, which can quantify the role of both dynamical variability and known thermodynamic aspects of climate change, and the interplay between them, in great spatio-temporal detail. As shown by Wehrli et al. (2020), the method can easily be expanded to a larger number of storylines for both past and future. The method could also be applied to other extreme events affected thermodynamically by climate change such as tropical cyclones (Feser and Barcikowska, 2012). Our future applications are, therefore, intended to cover a wide variety of extreme events over the historical record.

## 5. Code and Data Availability

The ECHAM6.1 global atmospheric model is available from the Max Planck Institute for Meteorology (MPI-M) website: https://mpimet.mpg.de/en/science/models/mpi-esm/echam/. The CMIP6 historical simulation data is archived at the World

Data Centre for Climate (WDCC): https://cera-www.dkrz.de/WDCC/ui/cerasearch/entry?acronym=RCM_CMIP6_Historical-HR. For analysis we have used the open access Python packages.

## 6. Author Contribution

LvG wrote the article, ran the simulations and analysed their results. FF and TS conceived the study and contributed to the writing and the interpretation of the results.

## 7. Competing Interest

The authors declare not to have any competing interests.

## 8. Acknowledgements

We would like to thank Dr Sebastian Rast from the Max Planck Institute for Meteorology (MPI-M) in Hamburg for his technical support applying spectral nudging in the ECHAM6 model, and Matthias Bittner and Wolfgang Müller from MPI-M

for providing ECHAM6 CMIP6 data. Simulations were carried out on the MISTRAL super computer at the German Climate Computing Centre (DKRZ), with technical support from Irina Fast. This work is a contribution to the "Helmholtz Climate Initiative REKLIM" (Regional Climate Change), a joint research project of the Helmholtz Association of German research centres (HGF), as well as to the European Research Council Advanced Grant "Understanding the atmospheric circulation response to climate change" (grant 339390). The authors would like to thank Francis Zwiers and an anonymous reviewer for

their thoughtful and constructive comments which helped improve the manuscript.

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

**Table 1. Greenhouse Gas concentrations for the ECHAM6 counterfactual simulations.**

| Greenhouse Gas | Concentration |
|---|---|
| Carbon dioxide ($CO_2$) | 285 ppmv |
| Methane ($CH_4$) | 790 ppbv |
| Nitrous oxide ($N_2O$) | 275 ppbv |
| Chlorofluorocarbons (CFC's) | 0 |


**Table 2. Example of attribution statements that are possible using the probabilistic and storyline approaches, for the case of the 2010 Russian heat wave.**

| | |
|---|---|
| **Probabilistic attribution** (based on results from Otto et al. (2012)) | Averaged over the Russian domain and over the month of July, temperatures in 2010 were 5C above the 1960s climatology, of which 4C was due to internal variability and 1C was due to anthropogenic climate change. |
| | The heatwave represented a 1-in-33 year event, which was three times more likely than it would have been in the 1960s. |
| **Storyline attribution** (based on present results) | Averaged over the Russian domain, temperatures in 2010 steadily increased from the 1985-2015 climatology through the month of July until about August 10, then rapidly returned to climatology. |
| | The domain-averaged heatwave reached 10C above the 1985-2015 climatology in early August, of which 8C was due to internal variability and 2C was due to anthropogenic climate change. |
| | The anthropogenic component of the warming reached 4C in the region to the south of Moscow during the first half of August, where it exacerbated the already warm temperatures there. |

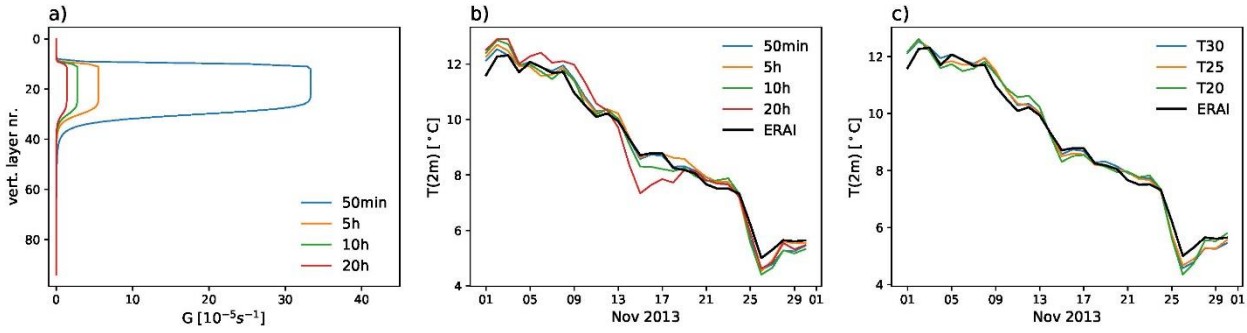

**Figure 1.** a) Nudging strength $G$ [$10^{-5}$ s$^{-1}$] as a function of model level, for different choices of minimum e-folding time as indicated. b) Daily mean temperatures at two-meter height [℃] of ECHAM6 in November 2013 averaged over the European domain (10°W-35°E/35-60°N) using the different e-folding times shown in panel a, in comparison to ERA-Interim. c) Daily mean temperatures as in panel b, but with a 50-minute nudging timescale at different truncations in comparison, again in comparison to ERA-Interim.

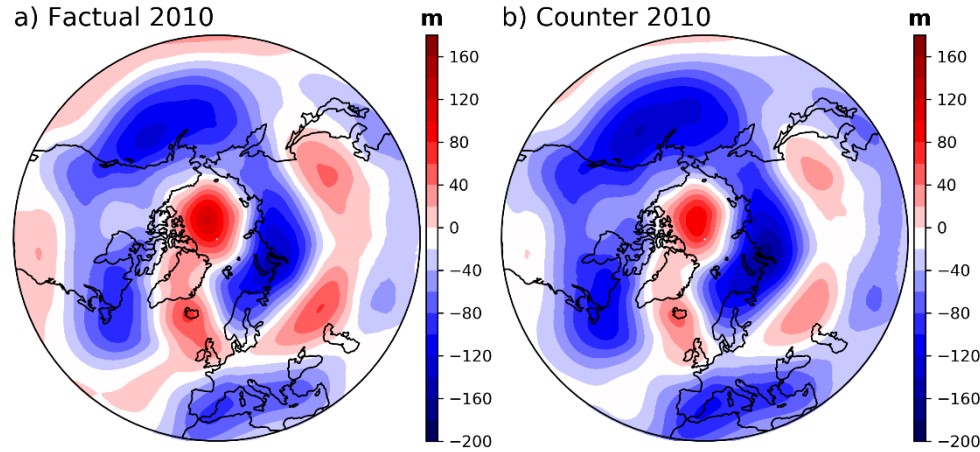

**Figure 2.** Geopotential height (z500) JJA anomalies [m] for the Northern Hemisphere showing the averaged spectrally nudged dynamic situation over a) factual members and b) counterfactual members of the summer 2010 blocking. Anomalies were calculated relative to the ECHAM_SN 1980-2014 JJA climatology.

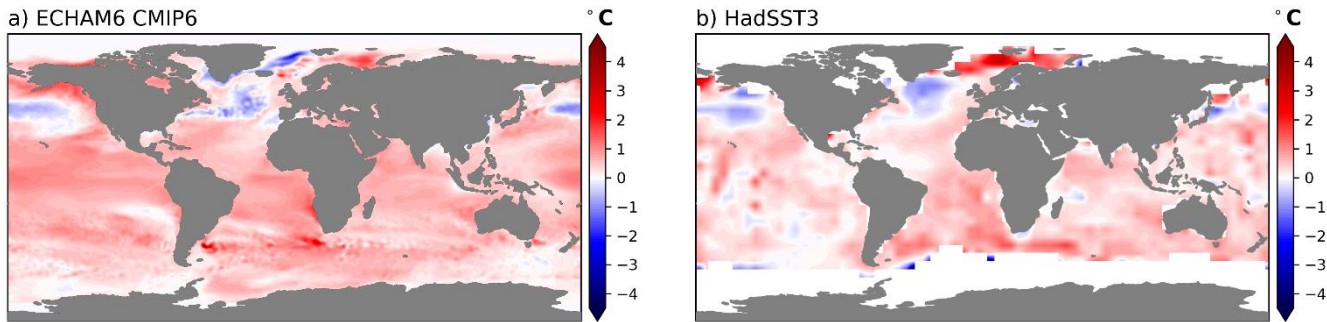

**Figure 3.** Sea Surface Temperature (SST) warming pattern [°C] a) calculated from ECHAM6 CMIP6 modelled data, and b) from HadSST3 observed data.

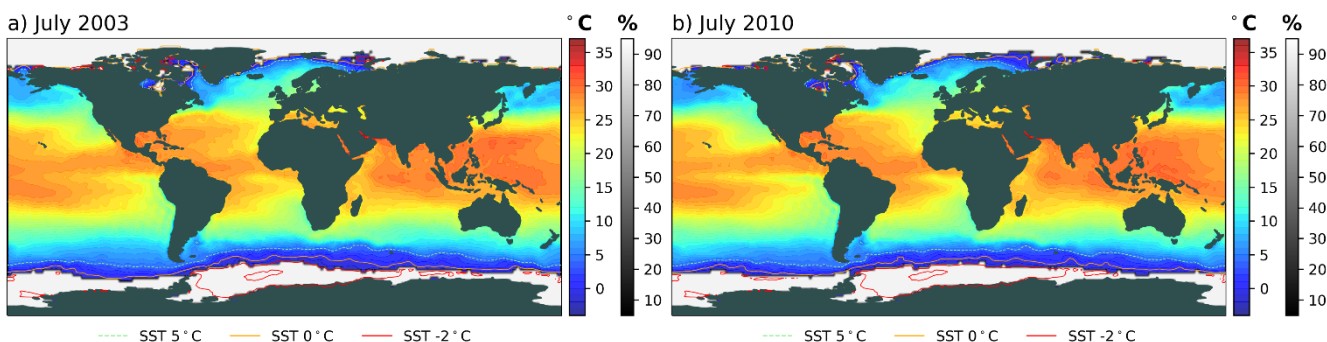

**Figure 4.** Counterfactual SST [°C] in shaded colours and factual SIC [%] in grayscale for (a) July 2003 and (b) July 2010. The SST 5°C (dashed green), 0°C (orange) and -2°C (red) contours are marked for reference.

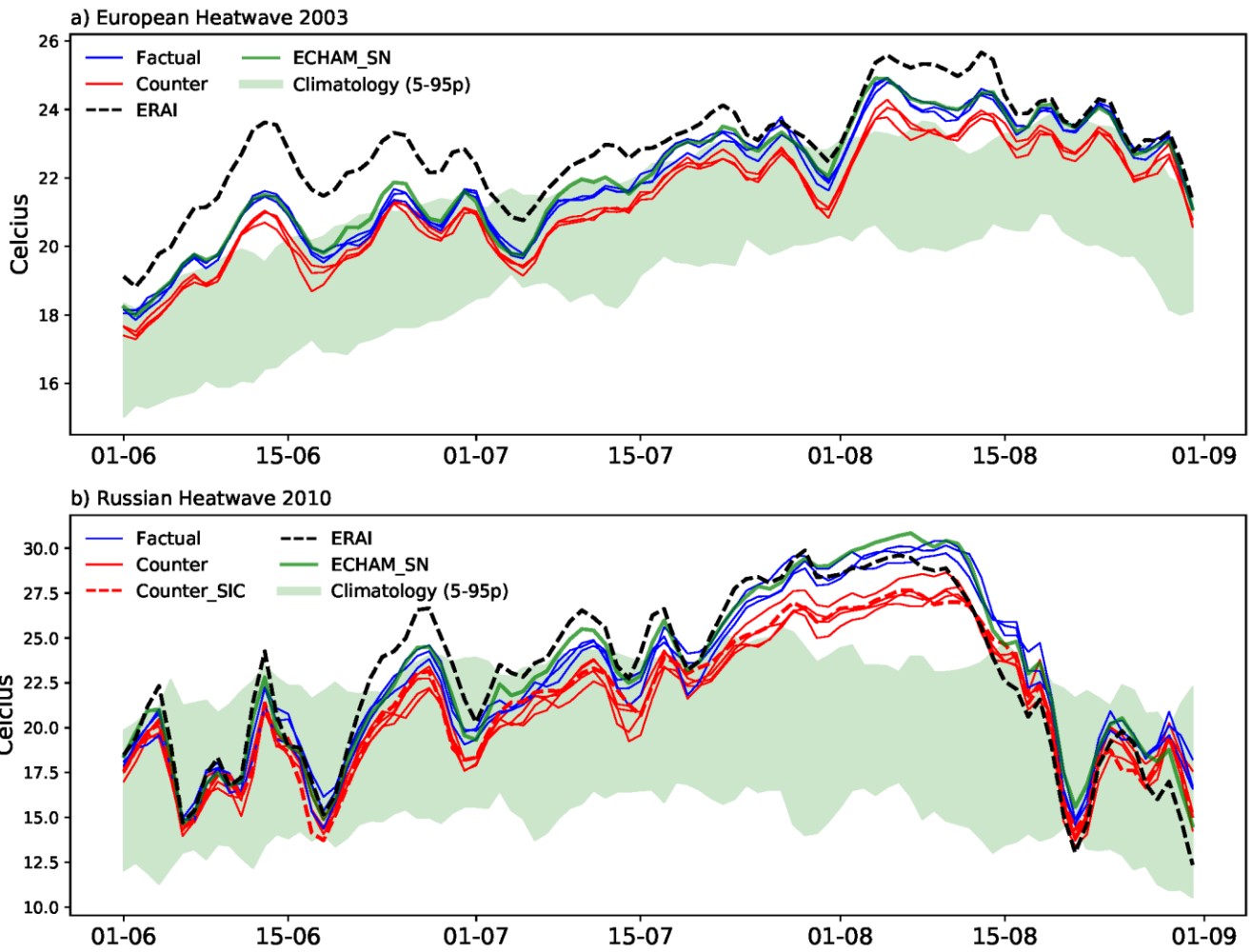

**Figure 5.** Daily mean temperature at two-meter height [°C] averaged over a) Europe (10°W-25°E/35-50°N) for summer 2003, and over b) Russia (35-55°E/50-60°N) for summer 2010, for the factual (blue), counterfactual (red), ECHAM_SN (green) simulations and ERA-Interim (dashed black) reanalysis data. The climatology (green shaded area) is the 5th-95th ranked percentile range between 1985-2015 calculated with ECHAM_SN (Schubert-Frisius et al., 2017). The dashed red line in panel b) shows the simulation with SIC changed in one of the counterfactual simulations (see text for details).

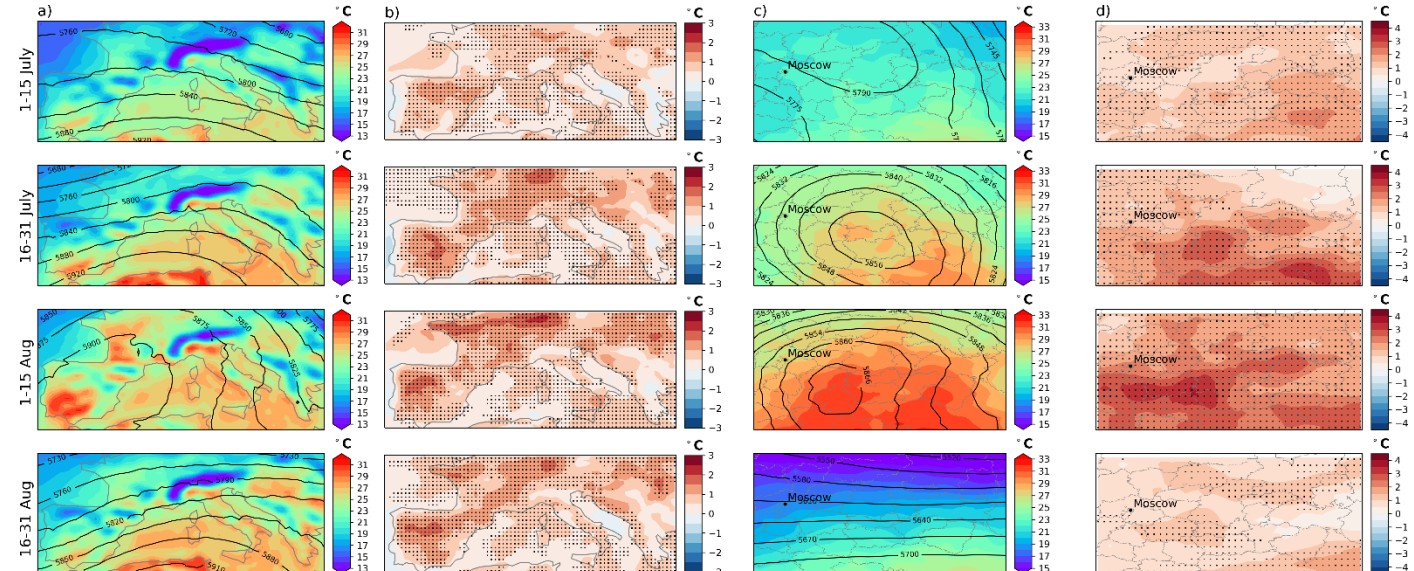

**Figure 6.** July and August divided into four half-month periods. Columns a and b show the European heatwave 2003, while columns c and d show the Russian heatwave 2010. In columns a and c, the factual geopotential height at z500 [m] is shown as black contour lines, while temperatures at two meters height [°C] are shown as shaded fields. In columns b and d, the differences in two-meter temperature [°C] between the factual and counterfactual simulations are shown as shaded fields. Stippling shows where all the factual members are >0.1°C above all the counterfactual members for that grid point. Note that the Russian domain is smaller, therefore the stippling has a different spacing than in the European domain.

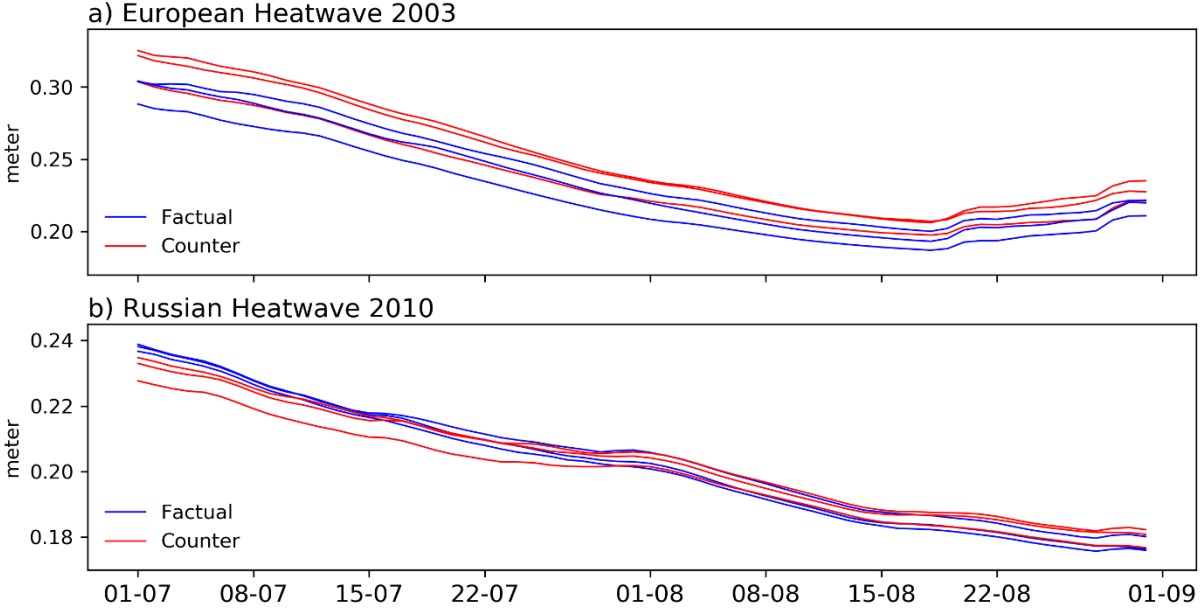

**Figure 7.** Average soil wetness in the root zone [m] averaged over Europe in 2003 and Russia in 2010, during July and August of each year. The factual simulations are shown in blue and the counterfactual simulations in red.

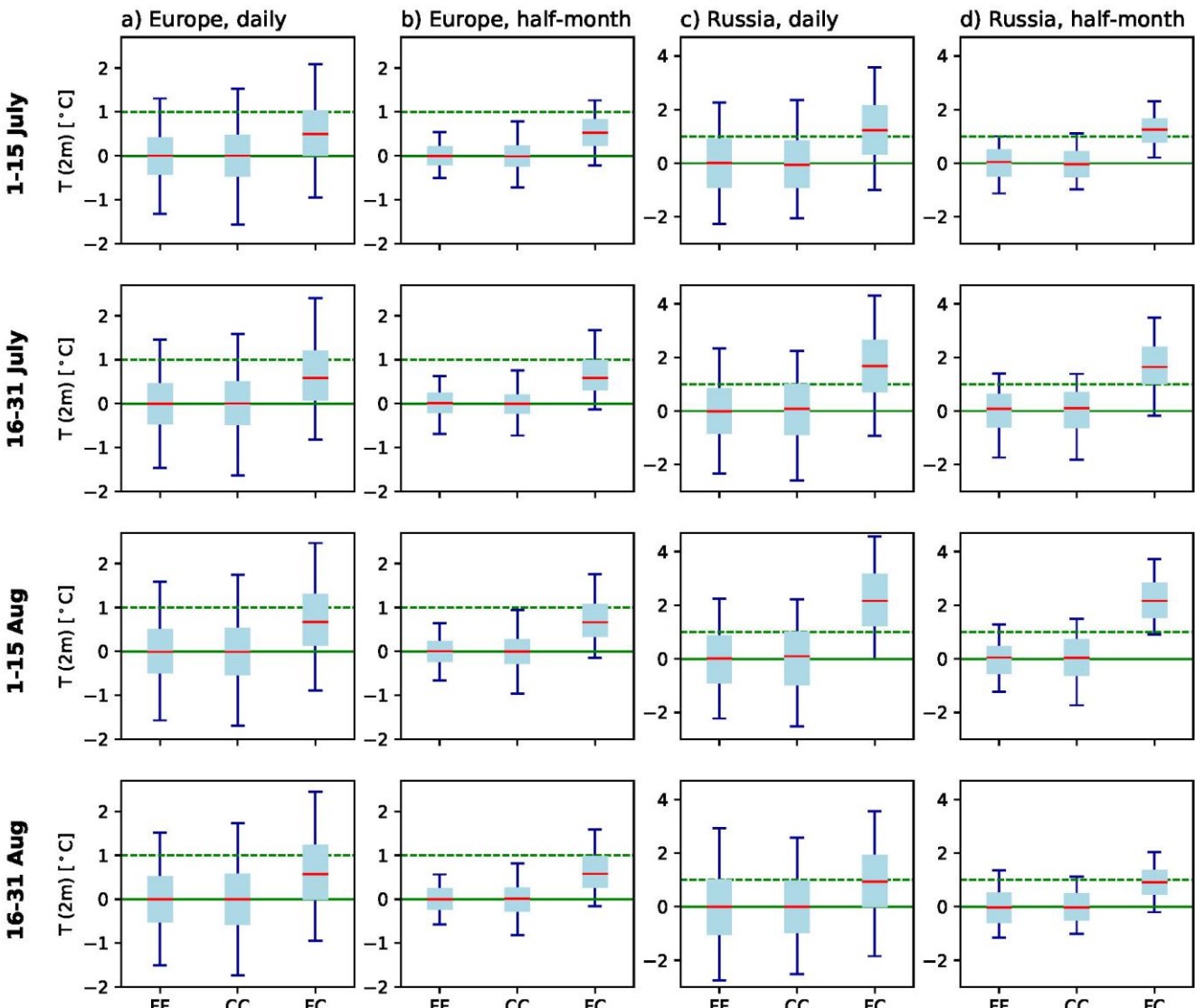

**Figure 8.** Distributions across grid points of differences between ensemble members in temperature at two meter height [°C], separated into the four half-monthly periods. FF = differences between pairs of factual members, CC = differences between pairs of counterfactual members, FC = differences between pairs of factual and counterfactual members. The boxes represent the 25th to 75th percentile range of the distributions, the red lines the 50th percentiles (the median), and the blue bars indicate the 5th to 95th percentile range. The dashed horizontal line indicates 1°C for reference. Columns a and b are for the European 2010 heatwave, and columns c and d for the Russian 2010 heatwave. Columns a and c show the differences of daily averages, and columns b and d the differences of half-monthly averages.