# Peer review of "A Methodology for Attributing the Role of Climate Change in Extreme Events: A Global Spectrally Nudged Storyline"

_Natural Hazards and Earth System Sciences, 2020_

## Referee Comment (RC1) · Francis Zwiers (Referee) · 18 Jul 2020

This paper applies the "storyline" approach to the analysis of the 2003 European and 2010 Russian heatwaves. The storyline approach to the analysis of extreme events is typically performed by constraining the circulation features in the factual and counter-factual representations of such events to be similar to features that were observed at the time of the event. In this case, the approach is to constrain the circulation within a global climate model to be similar to analysed circulation from a reanalysis via a spectral nudging technique.

General comments:

[Figure]

Even tighter constraints could, presumably, be obtained if a similar analysis were performed with a forecasting system that assimilated all observed data on the one hand (the factual case), and the same observed data except with "bogussed" temperatures on the other hand (the counterfactual case). While clearly not feasible (or expected) for this study, a similar study with a forecasting system might provide some additional useful insights into the application of storyline methods since the data that are presented to the model in the counterfactual case would then have to satisfy the thermal and dynamical balance constraints that would be imposed by the assimilation system. While this might make the counterfactual more difficult to implement, the use of an ensemble analysis and forecasting system would, in particular, provide some interesting possibilities for the quantification of uncertainties. Such an approach would also provide a "seamless" connection to probabilistic event attribution approaches (see next comment) that could draw on probabilistic weather forecasting techniques. Some discussion along these lines might be merited.

The introduction and the concluding discussion both try to make the case that the storyline approach is distinct from the probabilistic event attribution approach. I think, however, that the distinction is actually not very sharp. Rather, this is a question of conditional distributions and the degree of conditioning. The Stott et al., 2004, paper that started all of this off estimated distributions conditional on external forcing only (i.e., using a free running coupled model). Many subsequent papers estimated distributions conditional on external forcing and the pattern of sea-surface temperature anomalies that prevailed at the time of the event, largely because this enabled the production of very large ensembles of simulations with atmosphere-only models. In the storyline approach, conditioning is on external forcing, SST anomaly patterns, and circulation. In the case of this paper, a large-scale circulation constraint is applied globally via a spectral nudging approach. Even with this additional third constraint, the authors still, ultimately, end up trying to interpret the outcome in the context of uncertainty (e.g., by referencing estimates of climatological quantiles). Thus, even though they do not specifically estimate the factual and counterfactual distributions – interpretation becomes a statistical exercise. The fact that these distributions are not estimated reflects, I think, only a computational limitation (using an ensemble forecasting system in a parallel approach to the one taken in this paper would produce distributions that are conditional on the observed circulation). So, in my mind, this is not a matter of probabilistic vs non-probabilistic (or in medicine, epidemiological versus pathological) approaches to the interpretation of evidence, but rather simply a question of the degree of conditioning.

Some additional specific comments:

8: "are associated" –> "are often associated"

20-21: I suggest deleting this last sentence of the abstract. It isn't obvious how it follows from the preceding sentence, and also, there doesn't seem to be anything in the paper that discusses or explores this kind of application of the storyline methodology that is proposed.

43-47: I'm not sure that this view is as common as stated. I think what is understood is that large-scale internal variability is a feature of the dynamics (thermal and non-thermal) of the coupled Earth System, and that the dynamical changes tend not to be secular in the way that thermal changes are secular under external forcing (although there are a few exceptions – e.g., projections that storm tracks will shift a few degrees poleward, and the Southern Annular Mode response to stratospheric ozone forcing). Further, changes in vertical velocity are really hard to separate from purely thermal changes (despite some formalisms such as that of Bony et al., 2013) because of the feedbacks from latent heat release that are associated with a change in vertical motion.

77-78: I think it would be appropriate to mention Scinocca et al., 2016 (doi: 10.1175/JCLI-D-15-0161.1), who I think implemented a spectral nudging approach not dissimilar from the method used in this paper.

93: In this study the model is nudged towards reanalysis data, but in general, it could

be nudged to other types of data as well. For example, one might want to "dynamically downscale" a transient global climate change simulation with a much higher resolution global atmospheric model, nudging some aspects of the circulation of the high-resolution atmospheric model to that of the driving earth system model.

106-109: Notwithstanding the fact that there is probably not a lot of sensitivity to the choice of driving data (circulation is understood to be well-constrained by observations in reanalyses) it would still be useful to include some discussion of how the choice of driving data was made. Later, the paper makes some comparison between the nudged ECHAM6 output and ERA-Interim, so an immediate question might be, why not also use ERA-Interim (or perhaps better yet, ERA-5) as driving data. To the extent that ECHAM6 and ECMWF models still share common physics, there might also be an argument for using an ECMWF reanalysis product for driving ECHAM6 from a commonality of physics perspective.

160-162: I've always found the choice of counterfactual climate that is typically used in event attribution studies to be a bit unsatisfying. In effect, we need to trust that we can reliably adjust boundary conditions (such as SSTs) and reliably simulate a climate for which we have only very few observations. This choice allows a larger potential signal-to-noise ratio since it encompasses a relatively large amount of warming, but to the extent that it is important to have confidence that the counterfactual is well simulated, it might be preferable to use a period in the modern instrumental era when forcing was not as large.

171-172: I think it would be useful to say something about how well the large-scale circulation is constrained by the available observations. You've used NCEP1, but one could, for example, use an ensemble product such as the 20th century reanalysis (https://www.psl.noaa.gov/data/20thC_Rean/) to obtain an estimate of the strength of the observational constraint, at least in that product. The spread between ensemble members will be small for variables, periods and regions where the available observations provide effective constraints.

[Figure]

176: Formally at least, the quantity in brackets should also be a function of t rather than simply being fixed to a single number at each location (if nothing else, perhaps there is some seasonal variation in the pattern that would be relevant for the kind of short-term simulations used in this paper).

221: I would have thought that the IPCC AR5 Working Group I report would have been the best reference to cite to support a statement about how much warming has taken place.

235: "the hottest in the past..." –> "the hottest summer in the past ..."

236-238: As an aside, while these impacts, and those of the Russian heat wave described later, are large, they pale in comparison with the impacts that we are currently experiencing in the global pandemic.

241: "blocking" –> "block"

242: "by the atmospheric blocking" –> "by atmospheric blocking"

248: I think it is imperative to cite Stott et al., 2004, in this context as well.

254-264: It would be useful to compare the frequency of exceedance above the 95th percentile with what would be expected climatologically. We would expect exceedance to occur, on average, on 5% of days (that is, 4.5 days per season). Because of serial dependence, however, the expected interannual variability about that 4.5 day per season number is a bit difficult to calculate. Nevertheless, the counterfactual exceedance frequency would appear to be consistent with, or perhaps less than, the climatologically expected 4.5 days, whereas factual exceedance is clearly much higher than the expected frequency.

254-264 (Figure 5): Please include a curve for observed temperatures as well as the various simulated temperatures.

381: I think it would actually be useful to say a bit more about the noise level (there isn't

a lot on this aspect in the paper). In particular, the "noise level" reflects the variance of the temperature distribution after conditioning on the large scale circulation in the particular way that the conditioning has been done (the statistical interpretation is, ultimately, unavoidable, I think). If you change the constraint – for example, by changing aspects of the nudging strategy – then that "noise level" (aka, conditional variance) will change. I think readers should be made aware of those links and the impact that the study design choices could ultimately have on the attribution results that are obtained.

I hope you find these comments useful, and that everyone remains well in these unusual times.

Francis Zwiers

17 July 2020

---

## Referee Comment (RC2) · Anonymous Referee #2 · 20 Jul 2020

This paper proposes a new methodology based on global spectral nudging to perform extreme event attribution conditional to dynamical conditions, as part of a storyline approach of attribution. This method is applied to two case studies : the 2003 European heatwave and the 2010 Russian heatwave. I find the paper very clear and interesting and just have a few minor questions and comments for the authors that I list below.

l.57-59 I get your point about type 1 and type 2 error because I have read Lloyd and Oreskes' paper. However, I feel that this sentence does not fit very well in this paragraph and will be very confusing for someone who has not read the paper. I would delete the sentence or move it elsewhere and develop it a bit more.

[Figure]

l.205 Why did you choose a three years spin-up? How do you know this is enough?

l.209 Is there a reason behind the choice of three runs? Do you have any idea whether the results would be different if you added more runs? I understand that it takes computational time to add more runs for a global model but if you could comment on this (maybe as a limit of your study), the number of runs would look more justified.

l.213 Could you add a reference for this statement? I know you comment on this later in the paper, but I think you should put the reference here first.

l.266 to 279 I find this whole paragraph very interesting and original. Do you have an interpretation to explain the spatial variability of the differences between factual and counterfactual simulations?

l.311-313 that's an interesting interpretation. Do you have a reference about the direct radiative effect of GHG?

l.334 You could link that statement to the results presented in Hauser, M., Orth, R., and Seneviratne, S. I. (2016), Role of soil moisture versus recent climate change for the 2010 heat wave in western Russia, Geophys. Res. Lett., 43, 2819– 2826, doi:10.1002/2016GL068036.

---

## Author Response (AR1)

**Revision Document**

The significant changes are incorporated in the author's responses (underlined text)

P2-6             Authors reply on comments from Referee #1

P7-8             Authors reply on comments from Referee #2

P10              Paper with changes marked yellow

**Authors reply on comments from Referee #1**

*Many thanks to the reviewer for his time and effort to provide us with comments, they are very helpful. In the text below you will find our responses to each comment. The comments received concerning language are all accepted and changed accordingly in the main text; therefore, they are not further discussed.*

General comments: **Even tighter constraints could, presumably, be obtained if a similar analysis were performed with a forecasting system that assimilated all observed data on the one hand (the factual case), and the same observed data except with "bogussed" temperatures on the other hand (the counterfactual case). While clearly not feasible (or expected) for this study, a similar study with a forecasting system might provide some additional useful insights into the application of storyline methods since the data that are presented to the model in the counterfactual case would then have to satisfy the thermal and dynamical balance constraints that would be imposed by the assimilation system. While this might make the counterfactual more difficult to implement, the use of an ensemble analysis and forecasting system would, in particular, provide some interesting possibilities for the quantification of uncertainties. Such an approach would also provide a "seamless" connection to probabilistic event attribution approaches (see next comment) that could draw on probabilistic weather forecasting techniques. Some discussion along these lines might be merited.**

*Response: We agree that it would be useful to refer to the wider context of highly conditioned attribution and its potential connection to probabilistic NWP, and have added a brief discussion to that effect, referring to the 2016 US NAS report which discusses this prospect more fully than we are able to.*

Change made: L434-439: Elaborated on the continuum between storyline and probabilistic approaches and possible ways of implementing intermediate set-ups which would connect with probabilistic NWP.

**The introduction and the concluding discussion both try to make the case that the storyline approach is distinct from the probabilistic event attribution approach. I think, however, that the distinction is actually not very sharp. Rather, this is a question of conditional distributions and the degree of conditioning. The Stott et al., 2004, paper that started all of this off estimated distributions conditional on external forcing only (i.e., using a free running coupled model). Many subsequent papers estimated distributions conditional on external forcing and the pattern of sea-surface temperature anomalies that prevailed at the time of the event, largely because this enabled the production of very large ensembles of simulations with atmosphere-only models. In the storyline approach, conditioning is on external forcing, SST anomaly patterns, and circulation. In the case of this paper, a large-scale circulation constraint is applied globally via a spectral nudging approach. Even with this additional third constraint, the authors still, ultimately, end up trying to interpret the outcome in the context of uncertainty (e.g., by referencing estimates of climatological quantiles). Thus, even though they do not specifically estimate the factual and counterfactual distributions – interpretation becomes a statistical exercise. The fact that these distributions are not estimated reflects, I think, only a computational limitation (using an ensemble forecasting system in a parallel approach to the one taken in this paper would produce distributions that are conditional on the observed circulation). So, in my mind, this is not a matter of probabilistic vs non-probabilistic (or in medicine, epidemiological versus pathological) approaches to the interpretation of evidence, but rather simply a question of the degree of conditioning.**

*Response: We absolutely agree in principle with this comment, and have edited the text to avoid any misunderstanding. However, in practice, the difference in the sharpening of the pdfs that results from conditioning on SSTs and on circulation is enormous. It's perhaps analogous to NWP; in principle all NWP is probabilistic, but when the distribution is sharp (as it is for e.g. a stratospheric sudden warming a few days in advance, or a frontal passage 24 hours in advance) then the forecast is invariably interpreted deterministically. The probabilities arising from conditioning on SSTs have a natural physical interpretation in terms of seasonal predictability, but the probabilities arising from conditioning on circulation would not seem to be so easily interpretable. Thus we are using them here as uncertainties on our deterministic estimates, rather than as probabilistic predictions (for which an ensemble of three is anyway much too small). We have now acknowledged this limitation of our framework.*

Changes made: L57-62: Elaborated on the continuum between storyline and probabilistic approaches. L235-240: Acknowledged that our ensemble sizes are not large enough for our results to be interpreted as conditional probabilities, although this would be possible in principle.

Some additional specific comments:

**20-21: I suggest deleting this last sentence of the abstract. It isn't obvious how it follows from the preceding sentence, and also, there doesn't seem to be anything in the paper that discusses or explores this kind of application of the storyline methodology that is proposed.**

*Response: It is quite common for the last sentence of an abstract to discuss the potential implications of the findings of the paper, and we believe this particular sentence is well justified in that respect. In particular, the concept of a 'stress test' is very much a deterministic approach with no probability attached. We have expanded on the discussion to make this clear.*

No change made: On reflection, we feel that L421-422 sufficiently backs up this statement.

**43-47: I'm not sure that this view is as common as stated. I think what is understood is that large-scale internal variability is a feature of the dynamics (thermal and nonthermal) of the coupled Earth System, and that the dynamical changes tend not to be secular in the way that thermal changes are secular under external forcing (although there are a few exceptions – e.g., projections that storm tracks will shift a few degrees poleward, and the Southern Annular Mode response to stratospheric ozone forcing).**

*Response: We do not disagree with the statements made by the referee, but the cited text is not relevant to those points. That text is instead a discussion about how dynamical and thermodynamic components are identified in practice in diagnostic studies, and we believe that our discussion is representative of the state of the art in that respect. The points raised by the reviewer are, rather, relevant to the discussion immediately before. We have expanded on that discussion to reflect the reviewer's comments, referring to the results of Deser et al. (2016) who examined exactly this point for the case of temperature extremes.*

Change made: L40: Added reference to Deser et al. (2016)

**Further, changes in vertical velocity are really hard to separate from purely thermal changes (despite some formalisms such as that of Bony et al., 2013) because of the feedbacks from latent heat release that are associated with a change in vertical motion.**

*Response: We don't disagree, and are just referring to these methodologies as wider context, since they are used in practice. Our approach is not diagnostic, and should incorporate the sort of feedback that is mentioned by the reviewer. We have now highlighted this advantage over purely diagnostic approaches.*

Changes made: L44: Inserted the word "diagnostically" in our description of those methods.
L86-88: Made clear that our approach is physical and not diagnostic, so includes these feedbacks.

**77-78: I think it would be appropriate to mention Scinocca et al., 2016 (doi: 10.1175/JCLI-D-15-0161.1), who I think implemented a spectral nudging approach not dissimilar from the method used in this paper.**

*Response: We have added Scinocca et al., 2016 as reference.*

Change made: L85: Reference to Scinocca et al. (2016) added.

**93: In this study the model is nudged towards reanalysis data, but in general, it could be nudged to other types of data as well. For example, one might want to "dynamically downscale" a transient global climate change simulation with a much higher resolution global atmospheric model, nudging some aspects of the circulation of the high resolution atmospheric model to that of the driving earth system model.**

*Response: The application the reviewer suggests was actually the original motivation of spectral nudging, in von Storch et al. (2000), to which we refer, and the application to dynamical regional climate downscaling is mentioned explicitly in the reference to Feser and Barcikowska (2012). We have expanded slightly on the prospect of other applications at the end of the paper, but feel it is out of scope to go too far in this respect.*

Change made: L435-439: Discussed possible extensions of the approach.

**106-109: Notwithstanding the fact that there is probably not a lot of sensitivity to the choice of driving data (circulation is understood to be well-constrained by observations in reanalyses) it would still be useful to include some discussion of how the choice of driving data was made. Later, the paper makes some comparison between the nudged ECHAM6 output and ERA-Interim, so an immediate question might be, why not also use ERA-Interim (or perhaps better yet, ERA-5) as driving data. To the extent that ECHAM6 and ECMWF models still share common physics, there might also be an argument for using an ECMWF reanalysis product for driving ECHAM6 from a commonality of physics perspective.**

*Response: As the reviewer mentions, large-scale circulation is understood to be well-constrained by observations in reanalyses, so the choice of analysis product should not matter. In any case, the factual and counter-factual simulations are nudged to the same reanalysis, so any error in the reanalysis should cancel to a first approximation. We chose NCEP1 so that our method is applicable from 1948 onward as we are planning to use the method for a multi-decadal study, and NCEP has the longest time series. However, one could certainly use other reanalysis products for a nudging study; there is nothing in our methodology that is NCEP-specific. The reviewer may be correct that for certain kinds of extreme events where the dynamics and the thermodynamics are tightly connected (e.g. tropical cyclones), consistency of the physics would be an asset. We have edited our text to incorporate some of these points.*

Change made: L122-125: Stated the motivation behind choosing NCEP1 , and pointed out that consistency between the physics of the reanalysis and of the model could be beneficial for certain kinds of extreme events.

**160-162: I've always found the choice of counterfactual climate that is typically used in event attribution studies to be a bit unsatisfying. In effect, we need to trust that we can reliably adjust boundary conditions (such as SSTs) and reliably simulate a climate for which we have only very few observations. This choice allows a larger potential signal-to-noise ratio since it encompasses a relatively large amount of warming, but to the extent that it is important to have confidence that the counterfactual is well simulated, it might be preferable to use a period in the modern instrumental era when forcing was not as large.**

*Response: We mainly used this method for traceability with other studies. In the multi-decadal study that we are presently undertaking (see previous comment), we will indeed be examining the extent to which the inferred signal of climate change for smaller climate forcings (e.g. mid-century) is consistent with the observational SST changes since then. We have mentioned this as a potential way to check on the results.*

Change made: L180-184: Mentioned this as a possible check on the results when using our method.

**171-172: I think it would be useful to say something about how well the large-scale circulation is constrained by the available observations. You've used NCEP1, but one could, for example, use an ensemble product such as the 20th century reanalysis (https://www.psl.noaa.gov/data/20thC_Rean/) to obtain an estimate of the strength of the observational constraint, at least in that product. The spread between ensemble members will be small for variables, periods and regions where the available observations provide effective constraints.**

*Response: This would indeed be an interesting suggestion if one were interested in the first part of the 20$^{th}$ century (or even further back). However, as we would then be looking at the distribution, across reanalysis ensemble members, of the difference between the factual and the counter-factual simulations (rather than at the difference of the distributions), we expect the ensemble spread in the reanalysis would not make too much of a difference to the attribution of the anthropogenic effect, just increase the uncertainty in its estimation.*

Change made: L111-112: Explained why our method should be robust to the choice of reanalysis product.

**176: Formally at least, the quantity in brackets should also be a function of t rather than simply being fixed to a single number at each location (if nothing else, perhaps there is some seasonal variation in the pattern that would be relevant for the kind of short-term simulations used in this paper).**

*Response:* The reviewer is correct: the warming pattern would be more accurate if a function of season. This was not yet applied in our case. The warming pattern is the difference between the 2000-2009 historical SST values minus the preindustrial values. Because we are simulating a smaller amount of years there is no need to apply a weighting per year. For a longer simulation this should indeed be included as well.

Change made: L196-203: Corrected the warming pattern equation and explained the simplifications employed in this study.

**221: I would have thought that the IPCC AR5 Working Group I report would have been the best reference to cite to support a statement about how much warming has taken place.**

*Response: Agreed. We now cite the 2018 AR15 special report on global warming.*

Change made: L254: Added reference to IPCC (2018).

**236-238: As an aside, while these impacts, and those of the Russian heat wave described later, are large, they pale in comparison with the impacts that we are currently experiencing in the global pandemic.**

*Response: We completely agree.*

No change made.

**248: I think it is imperative to cite Stott et al., 2004, in this context as well**.

*Response: We have added Stott et al., 2004 as citation.*

Change made: L281: Added reference to Stott et al (2004).

**254-264: It would be useful to compare the frequency of exceedance above the 95th percentile with what would be expected climatologically. We would expect exceedance to occur, on average, on 5% of days (that is, 4.5 days per season). Because of serial dependence, however, the expected interannual variability about that 4.5 day per season number is a bit difficult to calculate. Nevertheless, the counterfactual exceedance frequency would appear to be consistent with, or perhaps less than, the climatologically expected 4.5 days, whereas factual exceedance is clearly much higher than the expected frequency**.

*Response: This is an interesting suggestion, but indeed there is a tremendous amount of serial correlation in these time series. In the case of the 2003 European heat wave, for example, the counter-factual is at the high end of the climatological distribution throughout the entire period. We do not see how we could do such a calculation in a defensible way, and prefer to just use the 95$^{th}$ percentile as a reference point, as we have done.*

No change made.

**254-264 (Figure 5): Please include a curve for observed temperatures as well as the various simulated temperatures.**

*Response: We have added ERA-Interim as a representation of the observed temperature to the figure. Although there is a time-dependent offset with our simulated factual temperatures, which is beyond the scope of this study to explore, this shows that the simulated temperatures are highly correlated with the observed temperatures.*

Changes made: Figure 5: Added ERA-I temperature.
L293-295 and 341-342: Discussed comparison between ERA-I and the factual simulations.

**381: I think it would actually be useful to say a bit more about the noise level (there isn't a lot on this aspect in the paper). In particular, the "noise level" reflects the variance of the temperature distribution after conditioning on the large scale circulation in the particular way that the conditioning has been done (the statistical interpretation is, ultimately, unavoidable, I think). If you change the constraint –**

**for example, by changing aspects of the nudging strategy – then that "noise level" (aka, conditional variance) will change. I think readers should be made aware of those links and the impact that the study design choices could ultimately have on the attribution results that are obtained.**

*Response: We agree, and this relates back to an earlier comment. Our noise level is conditional on our nudging strategy. We are only using it as a test of robustness of our inferred signal; we are not attempting to interpret the noise in a probabilistic manner, even though it presumably does have such an interpretation. We have edited the text to make these points.*

Changes made: L113-115: Explained our philosophy in determining a noise level.
L235-240: Acknowledged that our ensemble sizes are not large enough for our results to be interpreted as conditional probabilities, although this would be possible in principle.

**Authors reply on comments from Anonymous Referee #2**

*The authors thank the reviewer for the time and effort in providing us with comments, they are very helpful. In the text below you will find our responses to each comment.*

**I find the paper very clear and interesting and just have a few minor questions and comments for the authors that I list below.**

*Response: Thank you for your input and positive view on our paper. This is very much appreciated.*

**l.57-59 I get your point about type 1 and type 2 error because I have read Lloyd and Oreskes' paper. However, I feel that this sentence does not fit very well in this paragraph and will be very confusing for someone who has not read the paper. I would delete the sentence or move it elsewhere and develop it a bit more.**

*Response: We believe that the type 1/type 2 error issue is a major motivation for the storyline approach, so we feel that it needs to be mentioned here. We understand that the Lloyd & Oreskes paper might be a bit philosophical for some readers, so we have added a reference to Trenberth et al. 2015 who make this point in a more informal way.*

Change made: L66: Added reference to Trenberth et al. (2015).

**l.205 Why did you choose a three years spin-up? How do you know this is enough?**

*Response: We chose three years because it takes roughly three years for soil moisture to balance out towards the new normal (counterfactual), which is important for studying heat waves. We have tested the soil moisture levels in each of the spin-up years and found that between the second and third year there is almost no more difference found. We chose not to show this in the paper. We have added a sentence to the methodology section explaining that we found, through testing, the three year spin-up to be sufficient.*

Change made: L232-233: Explained why we chose a three year spin-up.

**l.209 Is there a reason behind the choice of three runs? Do you have any idea whether the results would be different if you added more runs? I understand that it takes computational time to add more runs for a global model but if you could comment on this (maybe as a limit of your study), the number of runs would look more justified.**

*Response: We agree that more explanation of why we chose a three-member ensemble would be helpful. It was done just to provide a first check on the robustness of our results, following the precedent of other studies. If the signal is already clear from three ensemble members, then that is enough. If three members are not enough, then the signal is anyway going to be small. Constructing a larger ensemble would use computational resources for no apparent gain, and we prefer to use our computational resources to look at longer time periods. Note that we already used the ECHAM_SN simulation as an out-of-sample test of the representativeness of our factual ensemble, and we have now performed a sensitivity test using altered SIC values which provides an out-of-sample test of the representativeness of our counter-factual ensemble. We have added some additional text to mention this.*

Change made: L113-115: Explained why we chose to use three-member ensembles.

L235-240: Discussed our out-of-sample tests on the robustness of our three-member ensembles.

**l.213 Could you add a reference for this statement? I know you comment on this later in the paper, but I think you should put the reference here first.**

*Response: We have added a reference to Wehrli et al., 2019 to support this claim.*

Change made: L246: Added reference to Wehrli et al. (2019).

**l.266 to 279 I find this whole paragraph very interesting and original. Do you have an interpretation to explain the spatial variability of the differences between factual and counterfactual simulations?**

*Response: Thank you. However, we do not have an interpretation of this variability, and prefer not to speculate.*

No change made.

**l.311-313 that's an interesting interpretation. Do you have a reference about the direct radiative effect of GHG?**

*Response: We have added the Wehrli et al., 2018 paper as a reference.*

Change made: L351: Added reference to Wehrli et al. 2019. In our response to the referee, Wehrli et al. 2018 was mentioned inadvertently.

**l.334 You could link that statement to the results presented in Hauser, M., Orth, R., and Seneviratne, S. I. (2016), Role of soil moisture versus recent climate change for the 2010 heat wave in western Russia, Geophys. Res. Lett., 43, 2819– 2826, doi:10.1002/2016GL068036.**

*Response: Thank you for bringing this paper to our attention. We have added a sentence to point out that our results are in agreement with the results of Hauser et at., 2016 despite a different methodology.*

Change made: L369-371: Compared the results of Hauser al. (2016) to our results.

[revised manuscript text omitted]